# Antitumoral Agent-Induced Constipation: A Systematic Review

**DOI:** 10.3390/cancers16010099

**Published:** 2023-12-24

**Authors:** Agnès Calsina-Berna, Jesús González-Barboteo, Silvia Llorens-Torromé, Joaquim Julià-Torras

**Affiliations:** 1Palliative Care Department, School of Medicine, The University of Vic—Central University of Catalonia (UVIC-UCC), 08500 Vic, Spain; 2Palliative Care Department, Institut Català d’Oncologia, 08908 Badalona, Spain; jjulia@iconcologia.net; 3Research and Knowledge Group in Palliative Care of Catalan Institute of Oncology (GRICOPAL), 08916 Badalona, Spain; jgonzalez@iconcologia.net (J.G.-B.); sllorens@iconcologia.net (S.L.-T.); 4Palliative Care Department, Institut Català d’Oncologia-L’Hospitalet, 08916 Badalona, Spain; 5School of Medicine and Health Sciences, Universitat Internacional de Catalunya, 08195 Sant Cugat del Vallès, Spain

**Keywords:** neoplasms, constipation, antineoplastic agents

## Abstract

**Simple Summary:**

Constipation is the third most common symptom in patients receiving cytotoxic chemotherapy, and it can impact their quality of life. There is no clear definition of constipation. Several factors may cause constipation in cancer patients, such as the direct effect of the tumor, comorbidities, polypharmacy, or antitumoral treatments. Being aware of its prevalence is crucial for all physicians and nurses who treat those patients to more precisely address the symptoms. A total of 22.07% of patients present constipation when they are under antitumoral treatments. The loss of enteric neurons following the administration of antitumoral treatments may cause constipation, but the etiology is not completely established. It is difficult to recommend specific treatments for antitumoral treatment-induced constipation without more knowledge of the specific cause. Narratives and scoping reviews recommend using laxatives as a treatment. Other studies show the benefits of acupuncture, moxibustion, osteopaths, and probiotics. In many countries, acupuncture and moxibustion are not available in the public system, and most healthcare professionals are not familiar with those methods. Thus, although laxatives may be used, more specific studies on the prevalence and etiology of and specific treatments for antitumoral treatment-induced constipation are needed.

**Abstract:**

Background: Constipation is a common symptom in patients receiving antitumoral treatment. The mechanisms underlying antitumoral agent-induced constipation (ATAIC) are poorly defined. This systematic review aimed to analyze and synthesize the available information related to the prevalence, etiology, and treatment of ATAIC. Methods: A systematic review following the Preferred Reporting Items for Systematic Reviews and Meta-Analysis (PRISMA) guidelines was conducted. The review included human studies written in English, French, or Spanish involving patients with cancer and containing information about the prevalence, etiology, and treatment of ATAIC. Results: A total of 73 articles were included. The reported prevalence ranged from 0.8% to 86.6%. Six studies reported an ATAIC prevalence of over 50%. The prevalence rates of constipation of grades 3 and 4 ranged between 0 and 11%. The importance of enteric neuronal integrity in gastrointestinal function was reported. The articles with the highest levels of evidence in relation to ATAIC treatment obtained in this systematic review studied treatments with acupuncture, sweet potato, osteopath, probiotics, and moxibustion. Conclusions: The prevalence of constipation in patients undergoing antitumoral treatment is very diverse. Studies specifically designed to report the prevalence of antineoplastic treatment-induced constipation are needed. The importance of enteric neuronal integrity in gastrointestinal function was described. Thus, neuroprotection could be an area of research for the treatment of chemotherapy-induced gastrointestinal disorders.

## 1. Introduction

Constipation is the third most common symptom in patients receiving cytotoxic chemotherapy. It ranges from 4.9 to 16% in outpatients receiving chemotherapy [1], and it is classified as severe in 5% and moderate in 11% of patients [2], with a great impact on the quality of life. Constipation may be a direct effect of the tumor [3], related to comorbidities [4], to collateral effects of the symptomatic treatment of cancer [5], or to its antitumoral treatment [2].

The mechanisms underlying antitumoral agent-induced constipation (ATAIC) are poorly defined [2]. ATAIC has been linked to specific chemotherapeutic agents such as thalidomide, cisplatin, and vinca alkaloids that induce true ATAIC in up to 80–90% of patients [2,6]. The invoked mechanism of ATAIC in this class of antitumoral agents is autonomic neuropathy [6] because of a direct effect of the alkaloid to the autonomic enteric neuronal system. Little is known about the mechanism of ATAIC related to other antitumoral agents.

Distinguishing true ATAIC from other causes of constipation is difficult, and it is a major issue hindering investigation [7]. Given the scarcity of the literature concerning ATAIC, it is hard to accurately estimate its prevalence and severity among all chemotherapy-treated persons with cancer.

In contrast to the high interest in addressing chemotherapy-induced nausea and vomiting [8], ATAIC has not received much attention in the literature, although there is clear evidence of the relation between common antitumoral agents and constipation [1].

In this context, this systematic review aimed to analyze and synthesize the available information related to the prevalence, etiology, and treatment of ATAIC.

## 2. Methods

### 2.1. Design and Data Sources

A systematic review was performed by following the Preferred Reporting Items for Systematic Reviews and Meta-Analysis (PRISMA) guidelines. This study project was registered at PROSPERO (registration #: CRD42023442306).

The search of the literature was conducted in the following databases: MEDLINE, PubMed, ISI Web of Science (ISI WoS), Cochrane Central Register of Controlled Trials (CENTRAL), and Scopus. The search included all publications in these databases from inception to June 2023.

### 2.2. Eligibility Criteria

All types of human studies (written in English, French, or Spanish) involving patients with cancer were eligible for inclusion. Studies containing information about the prevalence, etiology, and treatment of ATAIC in patients with cancer and published as full-text articles were included. Protocols of studies were excluded. No other restrictions were placed on study design or assessment measures. Studies on pediatric populations, studies on animals, and studies relating to constipation secondary to antiemetic treatment related to cancer treatment were excluded.

### 2.3. Search Strategy and Study Selection

A search strategy was designed by combining MeSH terms and free text. Table 1 shows the search strategy for the MEDLINE database, which was adapted for the other databases. Articles identified in the search were separately screened and assessed independently for inclusion by two reviewers using Covidence software. (A.C.-B. and J.J.-T. reviewed all articles).

The retrieved articles were sorted in three stages (by title, abstract, and full text) according to PRISMA guidelines [9] after removing duplicates.

Articles were excluded if they failed to meet the inclusion criteria; disagreements in this regard were separately solved by a third reviewer (JG-B).

Figure 1 illustrates the process of study selection and the reasons for exclusion.

### 2.4. Data Extraction, Data Analysis, and Synthesis of Findings

All included articles were divided into subgroups according to the predetermined classification related to the aims of this review: prevalence, etiology, and treatment of ATAIC. Then, the studies were analyzed by topic. Data for each topic were extracted from the studies.

The characteristics of each article, including author, country, year of publication, study design, and main content(s), are available in Table 2 (prevalence), Table 3 (etiology), and Table 4 (treatment).

The Grading of Recommendations, Assessment, Development and Evaluation (GRADE) [10] system was used to assess the certainty of the evidence of each outcome from the treatment results. The GRADE system includes five aspects: limitations, inconsistency, imprecision, indirectness, and publication bias. The certainty of the evidence was assessed as high, moderate, low, or very low by two independent reviewers (AC-B and JG-B) according to the GRADE approach.

## 3. Results

### 3.1. The Characteristics of Included Studies

A total of 73 articles—which included information on 41,146 patients with cancer—were included in the review. From those 73 articles, 54 were included as they mentioned data about the prevalence of ATAIC, 6 were for their information relating to the etiology of ATAIC, and 17 were included as they mentioned pharmacological or non-pharmacological treatments to alleviate ATAIC. Three articles were used for both prevalence and treatment purposes [11,12,13].

### 3.2. Prevalence

The 54 studies mentioning aspects related to prevalence included information on 33,409 patients with cancer. The reported prevalence rate ranged from 0.8% to 86.6%, and the mean prevalence rate considering all studies was 22.07%.

Six studies reported an ATAIC prevalence of over 50% [11,14,15,16,17,18]. Those six studies had higher reported results from patients treated with carboplatin in ovarian cancer [14]; with lenalidomide, thalidomide, and bortezomib in multiple myeloma [11]; with adjuvant chemotherapy in breast cancer [15,17]; and with chemotherapy in lung cancer [16,18].

The severity of constipation was reported in 13 of the 54 studies. The prevalence rates of constipation of grades 3 and 4 [19] ranged between 0 and 11%.

The antitumoral treatment was not mentioned in 28 studies; those studies referred to chemotherapy, without specifying which chemotherapy. From those who specifically mentioned any antitumoral treatment, the most frequent antitumoral treatments mentioned were carboplatin [14,20,21], oxaliplatin [13,22,23,24,25], bevacizumab [23,24], vinca alkaloids [14,26,27], and lenalidomide, thalidomide, and bortezomib [11,22,28,29].

Related to cancer diagnosis, 19 studies included patients with all kinds of cancer. Twelve studies mentioned the prevalence of constipation in patients with breast cancer [12,15,17,30,31,32,33,34,35,36,37,38], ten studies mentioned the prevalence of constipation in patients with lung cancer [16,18,21,35,37,39,40,41,42,43], six mentioned the prevalence in patients with gynecological cancer [14,20,32,36,37,44], four in patients with leukemia and lymphoma [27,29,30,45], three in patients with multiple myeloma [11,28,46], three with colorectal cancer [23,24,35], and four with another location [26,40,47,48].

### 3.3. Etiology

Regarding the six studies mentioning aspects related to etiology, two were case reports [49,50], one was a randomized controlled trial [51], and three were narrative reviews [2,52,53]. The total number of patients included was 145. One study focused on the treatment with 5-fluorouracil, one study used ipilimumab, study used vincristine, and in three studies, the antitumoral treatment was not specified.

The loss of enteric neurons following the administration of antitumoral treatments may cause constipation.

The main contents are available in Table 3.

### 3.4. Treatment

From the 17 studies mentioning aspects related to treatment, the number of patients included was 7786. From the type of cancers reviewed, five did not mention the location of the cancer [54,55,56,57,58], four mentioned treatments in patients with leukemia and lymphoma [6,27,59,60], three involved patients with breast cancer [12,61,62], two involved patients with multiple myeloma [11,63], two involved patients with lung cancer [18,64], and one involved patients with colorectal cancer [13].

From these 17 studies, 10 mentioned chemotherapies in general as antitumoral treatment [18,54,55,56,57,58,59,61,62,64]. The treatments that were specifically mentioned were vinca alkaloids [6,27,60], lenalidomide/thalidomide/bortezomib [11,63], and 5-fluorouracil/epirubicin/cyclophosphamide [12].

Six studies were randomized controlled trials [12,13,18,59,61,64], four were narrative reviews [11,54,55,63], three were systematic reviews [57,58,62], two were retrospective studies [27,60], one was a scoping review [56], and one was a case report [6].

In relation to the randomized controlled trials studied, two of them studied treatment with transcutaneous acupoint electrical stimulation [18,64], one studied auricular acupressure [61], one studied treatment with sweet potato [59], one studied osteopathy [12], and one studied probiotics [13].

The systematic reviews showed results related to acupuncture [57,58,62], probiotics, and moxibustion.

The scoping review mentioned high-fiber diets, fluid intake exercises, and laxatives [56].

The narrative reviews mentioned fluid intake, stool softeners, motility stimulants, neostigmine, and laxatives in general [11,54,55].

The retrospective studies reported the use of prophylactic laxatives and lubiprostone [27,60].

The case report explained the benefits of metoclopramide in a patient with ileus secondary to vincristine [6].

The main contents are explained in Table 4.

## 4. Discussion

Constipation is a feared and frequently reported symptom in patients with cancer treated with antitumoral agents. However, the results of this systematic review show the scarce knowledge of constipation related to the direct effects of antitumoral agents in the gastrointestinal tract that lead to constipation.

Pain has been the main symptom to address in cancer patients, but constipation is the third most common symptom in patients receiving cytotoxic chemotherapy [1], and it can impact the quality of life. There is no clear definition of constipation. Moreover, the characteristics of bowel function have been less assessed by physicians. Thus, historically, constipation is a symptom that has been less addressed. Patients may have a prior history of constipation before the cancer diagnosis, and it is crucial to analyze the causes of constipation. Constipation can appear at any moment of the cancer disease, but it is more severe as the illness becomes worse [65]. General recommendations include the use of non-pharmacological approaches and laxatives. As laxatives, a combination of stool softeners and stimulants is usually needed [11,55]. Laxatives can be administered while the patient is under a chemotherapy regimen.

### 4.1. Prevalence

Most of the studies that report the prevalence of constipation in this systematic review are not specifically designed to determine the prevalence of constipation and mention the prevalence of several symptoms. In addition, the definition of constipation is not even mentioned in most of them. As it appears in previous articles, some authors mention the need to improve the definition of constipation [66] as well as the need to consider other aspects that influence the presence of constipation in cancer patients. Thus, it is necessary to determine not only how often patients move their bowels, but also the characteristics of their stools as well as the side effects of laxatives or how much the quality of life is worsened when constipation appears [66].

Although most articles included in the study are in the period of 2007–2023, the search included all publications in databases from inception to June 2023.

The reported prevalence in this systematic review is very diverse and ranges from 0.8% to 86.6%. These results could be explained by the lack of a clear definition of ATAIC. The prevalence of constipation in patients with palliative care needs is also very diverse in the literature, ranging from 30 to 90% of patients [65]. Other studies in advanced patients mention prevalence rates of 43% [67]. It is noteworthy that only 13.1% of patients never described disturbed bowel function during their illness [66].

Only six studies mention prevalence rates greater than 50% [11,14,15,16,17,64]. The antineoplastic treatments used in those patients were carboplatin, thalidomide, and doxorubicin/cyclophosphamide, and in three studies, the antineoplastic treatment used is not specified.

As far as the severity of constipation is concerned, previous studies explain that the burden related to constipation is more severe as death approaches [65]. In this systematic review, the severity of constipation is only mentioned in 13 of the 54 studies [1,11,14,25,29,30,34,35,36,47,48,68,69].

The prevalence of grades 3 and 4 of severity is much lower than the prevalence of grades 1 and 2. The maximum prevalence of high-severity constipation (grades 3 and 4) was 11% in a study using thalidomide, melphalan, prednisone, and bortezomib.

Gender, polypharmacy—not only opioids—and other medical conditions have been clearly described as factors related to constipation [67]. Thus, to determine if constipation is more severe because of antitumoral treatments or due to other factors, studies should include the analysis of more variables, such as polypharmacy and chronic conditions.

### 4.2. Etiology

According to the results of this systematic review, the three main chemotherapeutic agents causing constipation are cisplatin, vinca alkaloids, and thalidomide. Thalidomide is currently less used than it was in the past. As far as cisplatin and vinca alkaloids are concerned, their effects on constipation could be related to the peripheral neurotoxicity of these treatments.

Only one of the studies mentioning aspects of etiology, a randomized clinical trial- [51], described that methanogenesis in patients with colorectal cancer under treatment with 5-fluorouracil had a positive association with constipation. The other studies mentioning aspects of etiology in this systematic review [2,49,50,52,53] only cited the importance of enteric neuronal integrity in gastrointestinal function. Thus, studies analyzing treatments involving neuroprotection could be designed as an initial step of personalized treatments for patients depending on the cause of constipation.

Currently, all patients follow the same general recommendations, such as the consumption of fiber or laxatives, without studying or considering each patient’s enteric neuronal system.

Studies performed on mice—which were not included in this review as they did not meet the inclusion criteria—described the effects of 5-fluorouracil (5-FU) on the gastrointestinal tract [70]. Those studies in mice show that 5-FU administered for 3 days increased gastrointestinal transit, induced acute intestinal inflammation, and reduced the proportion of neuronal nitric oxide synthase immunoreactive neurons. 5-FU administered for 14 days induced a delayed gastrointestinal transit, the inhibition of colonic migrating motor complexes, increased short and fragmented contractions, and myenteric neuronal loss [70].

More studies have studied the link of chemotherapy and the enteric neuronal system in mice. Vera performed a study in which cisplatin-treated mice had an enteric neuronal loss that was associated with reduced colonic motor activity and a shorter gastrointestinal transit time [71]. The authors concluded that chronic cisplatin treatment may induce an enteric neuropathy characterized by changes in myenteric neurons associated with marked gastrointestinal motor dysfunction. These promising studies in mice emphasize the importance of enteric neuronal integrity in gastrointestinal function and recommend more studies to be conducted on humans who receive those antineoplastic treatments.

### 4.3. Treatment

Seven out of the seventeen studies about the treatment of ATAIC have low or very low levels of evidence using the GRADE system [10].

Moreover, the treatments were not specifically focused on the etiology of this cause of constipation as it is not mentioned how the proposed treatment could benefit enteric neuropathy. This nonspecific approach of treating constipation without considering its etiology recalls how constipation is often treated. In this way, it reminds us of the situation of opioid-induced constipation, where the current guidelines still recommend laxatives as first-line therapy [72,73] instead of recommending the treatment with peripherally mu opioid receptor antagonists (PAMORAs). The use of PAMORAs could be the most reasonable approach focused on the mechanism of opioid-induced constipation. Some narrative reviews analyzed in this systematic review mentioned treatment with laxatives as a general recommendation for constipation instead of focusing on a more etiologic approach like the neuroprotection of the gut [55].

In 3 of these 17 studies related to treatment, vinca alkaloids were the antitumoral treatment used [6,27,60]. Vinca alkaloids are also the antineoplastic treatments that appear the most in studies of constipation in children with cancer [74]. From the rest of the 17 studies, 10 mention chemotherapy in general as a cause of constipation, and the others mention lenalidomide, thalidomide and bortezomib, vinca alkaloids, 5-fluorouracil epirrubicin, cyclophosphamide, and capecitabine plus oxaliplatin regimen.

The studies with the highest level of evidence retrieved in this systematic review mention treatment with transcutaneous acupoint electrical stimulation [18,64], auricular acupressure [61], acupuncture [62], sweet potato [59], osteopathy [12], probiotics [13,57], and moxibustion [58].

The use of complementary and alternative medicines in cancer patients to cope with ATAIC has been studied [75]. Toygar reported that 31.8% of patients knew about complementary and alternative medicine to cope with ATAIC [75]. The most common method was phytotherapy (97.1%), and the most used herb was apricot (39.4%). Individuals’ information sources on complementary and alternative medicine were mainly internet and social media instead of scientific sources or health professionals. Unfortunately, the study did not report the benefits of those treatments.

As far as acupuncture is concerned, studies in patients with opioid-induced constipation also reported benefits with acupuncture in a recent meta-analysis [76].

It should be considered that in many countries, acupuncture and moxibustion are not available in the public system, and most healthcare professionals are not familiar with those methods. Hence, although the results in this systematic review showed some effectiveness, most patients will not benefit from those treatments.

**Table 2 cancers-16-00099-t002:** Studies related to prevalence of ATAIC.

Author, Country, Year	Study Design and Methodology	Antitumoral Treatment	Diagnosis (Cancer Type)	Number of Patients Included	Prevalence of Constipation (%) and Comments
Buckinham et al., UK, 1997 [14]	Descriptive study	Carboplatin	Ovarian cancer	11	69.08%.
Sitzia et al., UK, 1997 [45]	Self-report questionnaire	Cyclophosphamide, doxorubicin, vincristine, and prednisolone	Non-Hodgkin’s lymphoma	19	47.48%
Williams et al., USA, 2006 [30]	Descriptive study	Chemotherapy	Leukemia, lymphomas, or breast cancer	26	Grade 1: 15.4% Grade 2: 26.9% Grade 3: 7.7% Grade 4: 7.7%
Janz et al., USA, 2007 [31]	Survey	Chemotherapy and hormonotherapy	Stage 0–II breast cancer	1372	30%
Widakowich et al., Belgium, 2007 [28]	Narrative review	Lapatinib in combination with paclitaxel	All cancers	Not mentioned	33%
Henry et al., USA, 2008 [77]	Cross-sectional survey	Chemotherapy and/or radiotherapy	All cancers	814	45%
Smith et al., USA, 2008 [11]	Narrative review	Lenalidomide, thalidomide, and bortezomib	Multiple myeloma	779	Lenalidomide: all grades, 39%; grade 3: 2% Thalidomide: all grades, 55%; grade 3: 8% Bortezomib: all grades, 42%; grade 3: 2%
Yamagishi et al., Japan, 2009 [1]	Cross sectional study	Chemotherapy	All cancers	462	Total: 16%; moderate: 11%; severe: 4.9%
Gay et al., Italy, 2010 [28]	Narrative review	Thalidomide, melphalan, prednisone, and bortezomib	Multiple myeloma	Not mentioned	Grades 3–4: 4–11%
Boussios et al., Greece, 2012 [22]	Narrative literature review	Oxaliplatin, imatinib, bortezomib, temsirolimus, sunitinib-sorafenib, and bevacizumab	All cancers	Not mentioned	Oxaliplatin: 32% Imatinib: 4–13% Bortezomib: 43% Temsirolimus: 20% Sunitinib–sorafenib: 34% Bevacizumab: 40%
Grenon et al., USA, 2013 [23]	Narrative review	Bevacizumab, fluorouracil, oxaliplatin, irinotecan, and aflibercept	Metastatic colorectal cancer	Not mentioned	Bevacizumab + irinotecan + fluorouracil: 4% Bevacizumab + 5-fluorouracil, oxaliplatin: 4% Aflibercept + 5-fluorouracil + irinotecan: 0.8%
Petterson et al., Sweden, 2013 [24]	Cross-sectional study	Capecitabine, oxaliplatin, folinic acid, bevacizumab, and irinotecan	Colorectal cancer	104	20%
Sanigaram et al., India, 2014 [32]	Prospective observational study	Chemotherapy	Breast cancer, cervical cancer and ovarian cancer	100	29%
Montemurro et al., Italy, 2015 [15]	Prospective, single-arm study	Adjuvant chemotherapy outside of a clinical trial	Breast cancer	604	52%
Rashid et al., USA, 2015 [33]	Retrospective cohort study	First-line treatment	Breast cancer	1682	14.2%
Saini et al., India, 2015 [39]	Observational prospective study	Chemotherapy	Breast cancer + lung cancer	174 (101 breast cancer + 73 lung cancer)	5-Fluorouracil + doxorubicin + cyclophosphamide: 6% 5-Fluorouracil + epirubicin + cyclophosphamide 1% Docetaxel 1% Paclitaxel 1%
Tachi et al., Japan, 2015 [34]	Survey	Chemotherapy	Breast cancer	48	Grade 1: 25.5% Grade 2: 6.4% Grade 3: 2.1%
Yeogh et al., Singapore, 2015 [78]	Retrospective study	Chemotherapy	All cancers	294	6.9%
Chopra et al., India, 2016 [79]	Prospective observational survey	Chemotherapy	All cancers	1008	20%
Colosia et al., USA, 2016 [48]	Systematic literature review	Pazopanib or another targeted cancer agent or cytotoxic chemotherapy	Metastatic soft tissue sarcoma	1013	Trabectidine weekly: Grade 1/2: 32% Grade 3/4: 2%
Wagland et al., UK, 2016 [69]	Survey	Chemotherapy	All cancers	363	59% Mild: 30% Moderate: 20% Severe: 9%
Daud et al., USA, 2017 [47]	Narrative review	Agents targeting the MAPK pathway	Metastatic melanoma	Not mentioned	Dabrafenib: any grade, 14%; grade 3/4: 2% Vemurafenib: any grade, 14%; grade 3/4: 2% Trametinib: any grade, 14%; grade 3/4: 0% Dabrafenib + trametinib: any grade, 22%; grade 3/4: 0%
Delforge et al., Austria, 2017 [46]	Case series	Proteasome inhibitors, immunomodulatory drugs, and monoclonal antibodies	Multiple myeloma	Not mentioned	Bortezomib iv: 2%; bortezomib sc: 1–2% Carfilzomib: 1% Ixazomib: <1% Thalidomide: 10% Lenalidomide: 2–3% Pomalidomide: 2% Grade 3/4: 2%
HSU et al., Taiwan, 2017 [20]	Survey	Carboplatin + taxol/topotecan or gemcitabine	Gynecological cancer patients	89	27.8%
Pearce et al., Australia, 2017 [35]	Prospective cohort study	Chemotherapy	Breast, lung, and colorectal cancers	449	74% (grade 1: 40%; grade 2: 25%; grade 3: 7%; grade 4: 2%)
Stansborough et al., Australia, 2017 [80]	Narrative review	Proteasome inhibitors	All cancers	Not mentioned	36.8%
Wahlang et al., India, 2017 [81]	Prospective observational study	Chemotherapy	All cancers	119	12.26%
Wong et al., USA, 2017 [16]	Cross-sectional study	Chemotherapy	Lung cancer	145	50.3%
Yang et al., China, 2017 [82]	Systematic review	Vandetanib	All cancers	6382	17%
Irwin et al., USA, 2018. [40]	Retrospective longitudinal observational cohort study	Ipilimumab, nivolumab, or ipilimumab/nivolumab combination therapy	Metastatic melanoma and lung cancer	2545	Lung cancer: ipilimumab: 19.1%; Nivolumab: 13.9%; combination therapy: 16.1% Melanoma: ipilimumab: 15%; nivolumab: 14.3%; combination therapy: 12.1%
Luo et al., China, 2018 [41]	Meta-analysis of randomized controlled trials	PD-1/PD-L1 inhibitors	Non-small-cell lung cancer	4413	PD-1, PDL-1 inhibitors: 6.34% Control (chemotherapy): 8.08%
Muthu et al., India, 2018 [42]	Prospective observational study	Chemotherapy	Lung cancer	112	20.1%
Lagrange et al., France, 2019 [12]	Single-left, randomized, double-blind, parallel, controlled clinical study	FEC 100-Taxotere chemotherapy	Breast cancer	94	Experimental group: 47.1%
Nyrop et al., USA, 2019 [17]	Incidence study	Chemotherapy	Breast cancer	152	Doxorubicin/cyclophosphamide plus paclitaxel: 47% Docetaxel/cyclophosphamide: 30% Docetaxel/carboplatin: 25% Doxorubicin/cyclophosphamide plus paclitaxel/carboplatin: 72%
Abdelhafeez et al., Egypt, 2020 [83]	Systematic review	Combination immune checkpoint inhibitors versus monotherapy	All cancer	2544	RR of constipation with combination: 8.4 RR all grades of adverse event; 1.48 high grade of adverse event.
Ataseven et al., Germany, 2020 [36]	Prospective interview study	Chemotherapy and targeted therapy with antibodies	Ovarian cancer and breast cancer	150	23%
Cheson et al., USA, 2020 [29]	Narrative review	Combination of rituximab and lenalidomide	Follicular and low-grade non-Hodgkin’s lymphoma	Not mentioned	Range: 26%, 27%, 35%, 45% Grade 3/4: 2%
Irfan et al., Pakistan, 2020 [26]	Retrospective study	Procarbazine, lomustine, and vincristine	Grade 2 oligodendroglioma, oligoastrocytoma, or astrocytoma	63	9.5%
Kawada et al., Japan, 2020 [27]	Retrospective cohort study	Vinca alkaloid	Acute lymphoblastic leukemia and malignant lymphoma	203	25%
Martin et al., USA, 2020 [44]	Qualitative research (interview)	Chemotherapy	Ovarian cancer	64	Patients from Europe: 4% Patients from USA: 3%
Saragioto et al., Brazil, 2020 [84]	Retrospective longitudinal study	Chemotherapy	All cancers	187	11.58%
Singh et al., USA, 2020 [37]	Survey	Chemotherapy	Breast, gastrointestinal, gynecological, and lung cancers	1251	43.2%
Thenmozhi et al., India, 2020 [85]	Cross-sectional study	Chemotherapy	All cancers	50	30%
Zorn et al., Germany, 2020 [86]	Cross-over trial	Chemotherapy	All cancers	30	8.8%
Chen et al., China, 2021 [21]	Cross-sectional study	Platinum	Lung cancer	100	41% Cisplatin: 26.8% Nedaplatin: 56.1% Carboplatin 17.1%
Crivelli et al., Italy, 2021 [68]	Survey	Chemotherapy	All cancer	283	In 2006–2007: grade 1: 48.6%; grade 2: 33.3%; grade 3: 2.8% In 2013–2014: grade 1: 40.1%; grade 2: 16.6%; grade 3: 0.5%
Huo et al., China, 2021 [87]	Meta-analysis	PD-1 inhibitors	Non-small-cell lung cancer	3716	2.8%
Mao et al., China, 2021 [18]	Randomized controlled trial	Chemotherapy	Lung cancer	122	Time point 1: Intervention group, 59%; control group, 55.7% (*p* = 0.71) Time point 2: Intervention group, 47.5%; control group, 65.6% (*p* = 0.045) Time point 3: Intervention group, 34.4%; control group, 68.9% (*p* = 0.000)
Duarte et al., Brazil, 2021 [88]	Diagnostic accuracy study	Chemotherapy	All cancers	240	86.6%
Orsi et al., Italy, 2021 [25]	Retrospective multileft study	Nab-paclitaxel plus gemcitabine; folinic acid, fluorouracil, irinotecan, and oxaliplatin cisplatin; nab-paclitaxel, capecitabine, and gemcitabine	Patients with cancer and documented germline pathogenic variants of BRCA1-2	85	Nab-paclitaxel plus gemcitabine: 11% grade 1–2; 0% grade 3; 0% grade 4 Folinic acid, fluorouracil, irinotecan, and oxaliplatin: 20% grades 1–2; 0% grade 3; 0% grade 4 Cisplatin, nab-paclitaxel, capecitabine, and gemcitabine: 3% grades 1–2; 0% grade 3; 0% grade 4
Yu et al., China, 2021 [43]	Observational study	Chemotherapy	Lung cancer	200	2.5%
Nguyen et al., Vietnam, 2022 [38]	Prospective study	Chemotherapy	Breast cancer	396	1.5%
Pehlivan et al., Turkey, 2022 [89]	Cross-sectional study	Chemotherapy	All cancers	252	31.7%
Huang et al., China, 2023 [13]	Randomized, single-blind, placebo-controlled prospective study	Capecitabine + oxaliplatin	All cancers	100	Patients under treatment with probiotic: 4% Patients under treatment with placebo: 28%

FEC 100: fluorouracil, epirubicin, and cyclophosphamide; RR: relative risk; Iv: intravenous; Sc: subcutaneous; RR: relative risk.

**Table 3 cancers-16-00099-t003:** Studies related to the pathophysiology of ATAIC.

Author, Country, Year	Study Design and Methodology	Antitumoral Treatment	Diagnosis (Cancer Type)	Number of Patients Included	Main Findings on Pathophysiology
Bhatia et al., USA, 2009 [49]	Case report	Ipilimumab	Melanoma	1	Biopsies of the colonic wall revealed prominent inflammatory infiltrates of mononuclear lymphocytes associated with the myenteric nervous system, consistent with immune-mediated inflammatory effects of ipililumab.
Leveque et al., France, 2009 [50]	Case report	Vincristine	Burkitt lymphoma	1	Ritonavir and lopinavir might have delayed vincristine elimination because CYP3A5 isoenzyme and glycoprotein P inhibition are involved in vincritine transportation and metabolism.
Holma et al., Finland, 2013 [51]	Randomized controlled trial	5-Fluorouracil	Colorectal cancer	143	Methanogenesis was a significant explaining factor with inverse association with diarrhea and positive association with constipation.
Stojanovsk et al., Australia, 2014 [52]	Narrative review	Not mentioned	All cancers	Not mentioned	Platinum-based chemotherapeutic agents could accumulate and enhance immune responses, and changes in neuroimmune interactions could possibly impact the gastrointestinal innervation and consequently cause long-term gut dysfunctions. Retention of reactive platinum compounds that are still capable of inducing DNA adducts can be found up to 20 years post-treatment with platinum-based agents.
McQuade et al., Australia, 2016 [2]	Narrative review	Not mentioned	All cancers	Not mentioned	Loss of enteric neurons following administration of cisplatin and oxaliplatin has been correlated with an increase in a population of the myenteric neurons expressing neuronal nitric oxide synthase. Studies emphasize the importance of enteric neuronal integrity in gastrointestinal function whilst suggesting neuroprotection as a potential therapeutic pathway for the treatment of chemotherapy-induced gastrointestinal disorders.
Escalante et al., Australia, 2017 [53]	Narrative review	Chemotherapy	All cancers	Not mentioned	Constipation could be due to neuronal loss limiting the innervation of the internal and external muscles of the gut, thus impairing normal colonic motor function, leading to constipation.

**Table 4 cancers-16-00099-t004:** Studies related to treatment of ATAIC.

Author, Country, Year	Study Design and Methodology	Antitumoral Treatment	Diagnosis (Cancer Type)	Number of Patients Included	Content and Main Findings	Quality of Evidence (Grade)
Carr et al., UK, 2008 [54]	Narrative review	Chemotherapy	All cancers	Not mentioned	Neostigime may have a role in the management of ileus.	Very low
Davila et al., USA, 2008 [55]	Narrative review	Chemotherapy	All cancers	Not mentioned	Laxatives can be used in the initial treatment of constipation. Stimulant laxatives (bisacodyl/senna) increase intestinal motor activity. If not effective, osmotic agents (lactulose/sorbitol) can be used. The use of drugs to improve colonic transit has been disappointing. Metoclopramide is ineffective. Tegaserod was removed from the market because of cardiovascular adverse effects.	Very low
Smith et al., USA, 2008 [11]	Narrative review	Lenalidomide, thalidomide, and bortezomib	Multiple myeloma	Not mentioned	Constipation Grade 1: Increase fluid intake + increase fiber intake + provide comfort, privacy + increase physical activity. Consider bowel regimen when constipating medications are prescribed (docusate 2–3 tablets per day, senna not to exceed 8 tablets a day). Constipation Grade 2: All of the grade 1 recommendations plus nutritional consultation + consider laxatives and stimulants: magnesium sulfate 15 g PO daily, magnesium citrate 200 mL PO daily, lactulose 15–60 mL PO daily, bisacodyl 5–20 mg PO at night or 10–20 mg rectally after a meal. Constipation Grade 3: All of the grade 2 recommendations plus initiate bowel regimen + assess for bowel obstruction + consider IV hydration. If no response, consider referral to a gastroenterologist. Grade 4: All of the grade 3 recommendations plus hospitalization + rule out perforation.	Very low
Shin et al., Korea, 2016 [61]	Randomized controlled trial	Chemotherapy	Breast cancer	52	Patients treated with auricular acupressure experimented lower scores of constipations compared to usual care (*p* < 0.001). Patient Assessment of Constipation–Quality of Life scores of the experimental group were significantly lower compared with the control group (*p* < 0.001).	High
Zou et al., China, 2016 [59]	Randomized controlled trial	Chemotherapy	Leukemia	120 (57 group control + 63 group intervention)	Patients that eat sweet potato (200 g/day) significantly improved some items regarding defecation in Rome III criteria compared to routine nursing methods.	High
Fox et al., Ireland, 2017 [56]	Scoping review	Chemotherapy	All cancers	25 in Chou et al. (1 of the 27 articles from the review)	High-fiber diet, increase fluid intake (eight 8-ounce glasses of fluid/day), exercise, laxatives. Benefit not mentioned (it is a recommendation).	Moderate
Hayashi et al., Japan, 2017 [60]	Retrospective study	Vincristine-based chemotherapy regimen	Non-Hodgkin’s lymphoma	211	The incidence of constipation did not significantly differ between patients who received and did not receive prophylactic laxatives (30.2% versus 37.6%, respectively, *p* = 0.269). Magnesium oxide at doses of <2000 mg/d was not significantly effective for prevention of constipation, although the compound completely inhibited the incidence of constipation at doses of >2000 mg/d.	Low
Paner et al., USA, 2018 [63]	Narrative review	Bortezomib, lenalidomide, and dexamethasone	Multiple myeloma	Not mentioned	The authors recommend adding stool softeners and motility stimulant to relieve constipation if increasing oral fluid intake and dietary fiber (20–25 g/day) are insufficient.	Very low
Lagrange A et al., France, 2019 [12]	Randomized controlled trial	FEC 100-Taxotere chemotherapy	Breast cancer	94	Osteopath (experimental group) for a 15 min session. A 15 min session of osteopath did not show significant differences against control group concerning the rate of constipation (*p* = 0.204) according to clinician-reported side effects, but patients reported that the impact of constipation on quality of life was significantly lower in experimental group (*p* = 0.036).	High
Masoumi et al., Iran, 2019 [6]	Case report	Vincristine	Acute lymphoblastic leukemia	1	Metoclopramide can be considered for ileus treatment after ruling out the possibility of bowel obstruction. Prophylactic stool softeners should be administered to all patients undergoing chemotherapy with vincristine to prevent gastrointestinal motility disorders.	Very low
Kawada et al., Japan, 2020 [27]	Retrospective study	Vinca alkaloid	Acute lymphoblastic leukemia and malignant lymphoma	203	Patients treated with lubiprostone were significantly less likely to experience intractable constipation than those treated with stimulant laxatives (10% vs. 34%, *p* = 0.002).	Low
Chan et al., China, 2021 [62]	Systematic review	Chemotherapy	Breast cancer	1189	Acupuncture (penetrating needles on the acupoints) showed beneficial effects on constipation.	High
Mao et al., China, 2021 [18]	Randomized controlled trial	Chemotherapy	Lung cancer	122	Acupoint stimulation. Transcutaneous acupoint electric stimulation and gastric electrical stimulation were performed for 25 min daily for 14 days. Prevalence rates of constipation: 1 day before chemotherapy: Intervention group, 59%; control group, 55.7% (*p* = 0.71). 14 days after chemotherapy: Intervention group, 47.5%; control group, 65.6% (*p* = 0.045). 28 days after chemotherapy: Intervention group, 34.4%; control group, 68.9% (*p* = 0.000). At time points 2 and 3, constipation in the stimulation group had statistically significantly improved compared with the control group (*p* < 0.05).	High
Mao et al., China, 2021 [64]	Randomized controlled trial	Chemotherapy	Non-small-cell lung cancer	60	Transcutaneous acupoint electrical stimulation (TAES) was effective for alleviating constipation. TAES was provided daily for 30 min for 4 weeks, six times per week. BSFS and the CAS scores were used. Both scores for the experimental group were significantly higher than those for the control group (*p* = 0.004 and *p* < 0.001).	High
Garczyk et al., Poland, 2022 [57]	Systematic review	Chemotherapy	All cancers	2619	Two studies showed improvement of constipation with probiotics. Decrease in the duration of constipation and less severe ailments were noted using compositions containing L. acidophilus, B. infantis, and L. rhamnosus.	High
Yao et al., China, 2022 [58]	Systematic review	Chemotherapy	All cancers	2990	Compared to the controls, moxibustion significantly reduced the incidences of constipation (RR 0.59, 95% CI 0.44–0.78). Quality of life scores significantly improved after moxibustion. All adverse events related to moxibustion were mild.	High
Huang et al., China, 2023 [13]	Randomized controlled trial	Capecitabine + oxaliplatin regimen	Colorectal cancer	100	Treatment with probiotics proved to be significatively effective in the treatment of constipation when compared to placebo (probiotics: 4% constipation; placebo: 28% constipation (*p* = 0.019)).	High

g: grams; PO: oral route; ml: milliliters; mg: milligrams; d: day; min: minutes; BSFS: the Bristol Stool Form Scale; CAS: Constipation Assessment Scale. GRADE Working Group grades of evidence. High quality: Further research is very unlikely to change our confidence in the estimate of effect. Moderate quality: Further research is likely to have an important impact on our confidence in the estimate of effect and may change the estimate. Low quality: Further research is very likely to have an important impact on our confidence in the estimate of effect and is likely to change the estimate. Very low quality: We are very uncertain about the estimate.

## 5. Conclusions

The prevalence of constipation in patients undergoing antitumoral treatment is very diverse. It is important to emphasize the need to assess ATAIC in all cancer patients. Studies specifically designed to report the definition and prevalence of ATAIC are needed.

A specific pathophysiological approach to personalize the treatment of ATAIC is urgently needed. The importance of enteric neuronal integrity in gastrointestinal function has been described in basic research. Thus, future research should be focused on the effect of ATAIC on enteric neurons. Neuroprotection could be an area of research for the treatment of ATAIC. As far as treatment is concerned, research should be focused on the specific causes of constipation, avoiding general recommendations for all patients. Therefore, a more personalized approach will be performed if all patients are assessed in terms of their prevalence, their causes of constipation, and their specific treatments. Thus, although laxatives may be used, more specific studies on the prevalence and etiology of and specific treatments for antitumoral treatment-induced constipation are needed.

### The Strengths and Limitations of This Study

To the best of our knowledge, this is the only systematic review of ATAIC in adults. It includes articles from four databases and does not exclude any articles for methodological reasons.

Important clinical issues regarding ATAIC such as prevalence, etiology, and treatment have been addressed in this systematic review to provide useful information for clinicians.

However, this study presents some limitations which should be recognized. First, there was a huge amount of methodological heterogeneity among the included studies that limited the strength of some of the final conclusions.

Second, studies mentioning etiology report hypotheses and do not report strong conclusions about the mechanism by which ATAIC occurs.

Finally, studies reporting treatment options are few and not oriented to the presumable cause of ATAIC. None of the studies related to treatment are focused on treatments that could enhance the neuroprotection of the gut.

## Figures and Tables

**Figure 1 cancers-16-00099-f001:**
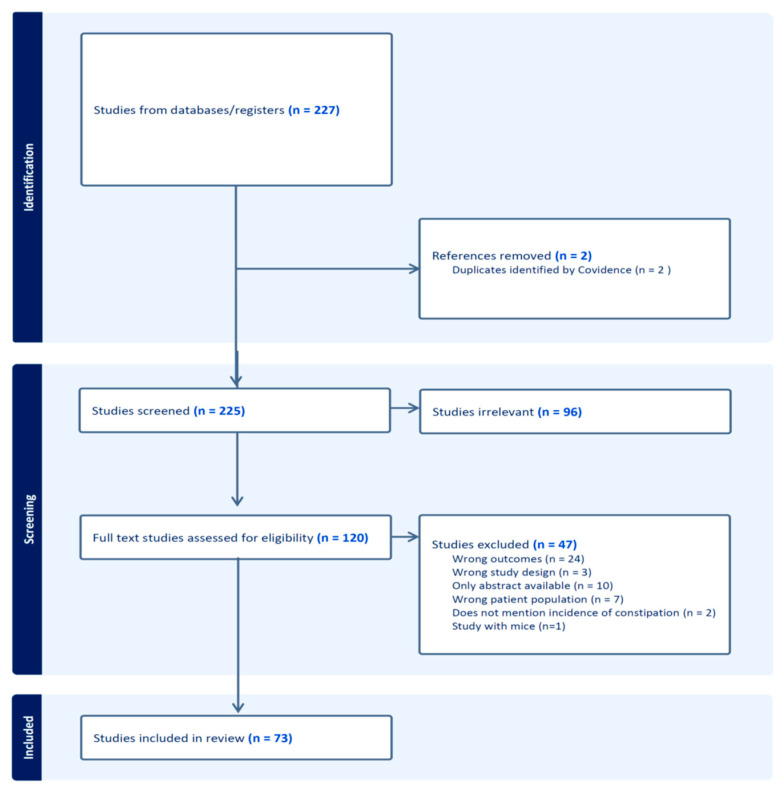
Preferred Reporting Items for Systematic Reviews and Meta-Analysis flow diagram of study selection.

**Table 1 cancers-16-00099-t001:** Search strategy.

Concept	Query Number	Query Content
	#10	#3 AND #6 AND #9 AND
	#9	#7 OR #8
Treatment	#8	“Antineoplastic Agents” [Mesh]
	#7	“Anticancer Agent*” OR “Antineoplastic Drug*” OR “Antitumor Drug*” OR Antineoplastic* OR Chemotherapy* OR Immunotherapy [Mesh]* OR Hormonotherapy*
	#6	#5 NOT #4
Problem	#5	Constipation [MESH]
	#4	Opioid-Induced Constipation [MESH]
	#3	#1 OR #2
Population	#2	Neoplasms [MESH]
	#1	Tumor* OR Tumor* OR Malign* Or Cancer* OR Neoplasms* OR Carcinoma* OR Oncol*

## Data Availability

Data are contained within the article.

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
