# Peer review of "Antitumoral Agent-Induced Constipation: A Systematic Review"

_cancers, 2023, doi:10.3390/cancers16010099_

Round 1
Reviewer 1 Report
Comments and Suggestions for Authors
The review draws attention to an important symptom in patients treated with anti-tumor drugs - the appearance of constipation. A summary of the prevalence of constipation due to anti-tumor treatment in patients is presented.
The authors found that the prevalence of ATAIC is very variable and further studies are needed. The mechanism of occurrence of ATAIC has not yet been established, only some assumptions have been made. The studies that describe the treatment of ATAIC are very few and do not offer a solution to the problem.
1. To improve the manuscript, I would recommend that the authors describe the period covered in the study. Most articles included in the study are in the period 2007-2023. Does this fact mean that the problem did not exist before or there are no studies from past years?
2. In tables 2-4, when indicating the authors, there should be a citation number for easier finding of the relevant article.
3. The authors have read a large number of articles on the subject. I would recommend in the Conclusion section to re-emphasize the importance of ATAIC and make some recommendations for future research and treatment.
Comments on the Quality of English Language
Minor editing of English is required
Author Response
Dear reviewers, we appreciate your comments as they improve the manuscript. The changes performed in the text are as follows:
|
Reviewer 1 |
|
|
I would recommend that the authors describe the period covered in the study. Most articles included in the study are in the period 2007-2023. Does this fact mean that the problem did not exist before or there are no studies from past years? |
Regarding the period covered in the study, as it is mentioned in the text: “The search included all publications in these databases from inception through June 2023”. The problem existed before but the lack of a clear definition makes it more difficult to establish a prevalence and physicians in the past probably paid more attention to symptoms such as pain rather than constipation. |
|
In tables 2-4, when indicating the authors, there should be a citation number for easier finding of the relevant article: |
It has been added.
|
|
The authors have read many articles on the subject. I would recommend in the Conclusion section to re-emphasize the importance of ATAIC and make some recommendations for future research and treatment.
|
Conclusions have been modified as follows: “The prevalence of constipation in patients with antitumoral treatment is very diverse. It is important to emphasize the need to assess ATAIC in all cancer patients. Studies specifically designed to report the definition and prevalence of ATAIC are needed. A specific pathophysiological approach to personalize the treatment of ATAIC is urgently needed. The importance of enteric neuronal integrity in gastrointestinal function has been described in basic research. Thus, future research should be focused on the effect of ATAIC on the enteric neurons. Neuroprotection could be an area of research approach for the treatment of ATAIC. As far as treatment is concerned should be focused on the specific causes of constipation, avoiding general recommendations for all patients. Therefore, a more personalized approach will be performed if all patients are assessed in their prevalence, their causes of constipation, and their specific treatment. Thus, although laxatives may be used, more specific studies on prevalence, etiology, and specific treatments for antitumoral treatments-induced constipation are needed.” |
Reviewer 2 Report
Comments and Suggestions for Authors
The discussion should be revised and improved. My suggestions are the following:
- According to your results, the three main chemotherapeutic agents causing constipation are cisplatin, vinca alkaloids, and thalidomide. How could it be explained? Is there any mechanism of action in common with those drugs that generate constipation in patients?
- When carrying out this review in search of the prevalence, etiology, type of cancer, and presence of constipation, what is the reason why this possible effect of chemotherapy in patients is not so reported or not so visible? Is the discomfort in patients less significant than other effects? Do doctors give less importance to it? Do patients have a history of suffering from constipation? Does the constipation start at the first chemo cycle?
- What are the best medications to relieve constipation? Do they have any interaction with chemotherapy agents?
A general discussion regarding the most successful constipation treatments should be included
Author Response
Dear reviewers, we appreciate your comments as they improve the manuscript. The changes performed in the text are as follows:
|
Reviewer 2: |
|
|
According to your results, the three main chemotherapeutic agents causing constipation are cisplatin, vinca alkaloids, and thalidomide. How could it be explained? Is there any mechanism of action in common with those drugs that generate constipation in patients? |
A paragraph in the discussion section has been added: “According to the results of this systematic review, the three main chemotherapeutic agents causing constipation are cisplatin, vinca alkaloids, and thalidomide. Thalidomide is currently less used than it was in the past. As far as cisplatin and vinca alkaloids are concerned, their effects on constipation could be related to the peripheral neurotoxicity of these treatments. “ |
|
When carrying out this review in search of the prevalence, etiology, type of cancer, and presence of constipation, what is the reason why this possible effect of chemotherapy in patients is not so reported or not so visible? Is the discomfort in patients less significant than other effects? Do doctors give less importance to it? Do patients have a history of suffering from constipation? Does the constipation start at the first chemo cycle? A general discussion regarding the most successful constipation treatments should be included.
|
Some paragraphs in the discussion and conclusions section have been added to address those issues: Discussion: “Pain has been the main symptom to address in cancer patients, but constipation is the third most common symptom in patients receiving cytotoxic chemotherapy [1] and it can impact the quality of life. There is no clear definition of constipation. Moreover, the characteristics of bowel function have been less assessed by physicians. Thus, historically has been a symptom less addressed. Patients may have a prior history of constipation before the cancer diagnosis, and it is crucial to analyze the causes of constipation. Constipation can appear at any moment of the cancer disease, but it is more severe as the illness is worse [67]. General recommendations include non-pharmacological approaches and laxatives. As laxatives, a combination of stool softeners and stimulants is usually needed [11, 56]. Laxatives can be administered although the patient is under a chemotherapy regimen.” Conclusions: “The prevalence of constipation in patients with antitumoral treatment is very diverse. It is important to emphasize the need to assess ATAIC in all cancer patients. Studies specifically designed to report the definition and prevalence of ATAIC are needed. A specific pathophysiological approach to personalize the treatment of ATAIC is urgently needed. The importance of enteric neuronal integrity in gastrointestinal function has been described in basic research. Thus, future research should be focused on the effect of ATAIC on the enteric neurons. Neuroprotection could be an area of research approach for the treatment of ATAIC. As far as treatment is concerned should be focused on the specific causes of constipation, avoiding general recommendations for all patients. Therefore, a more personalized approach will be performed if all patients are assessed in their prevalence, their causes of constipation, and their specific treatment. Thus, although laxatives may be used, more specific studies on prevalence, etiology, and specific treatments for antitumoral treatments-induced constipation are needed.” |